# The Tower Building Task: A Behavioral Tool to Evaluate Recreational Risk-Taking

**DOI:** 10.3390/bs12090325

**Published:** 2022-09-08

**Authors:** Santiago Gracia-Garrido, Marcos F. Rosetti, Kevin Muñoz Navarrete, Robyn Hudson

**Affiliations:** 1Instituto de Investigaciones Biomédicas, Universidad Nacional Autónoma de México, Mexico City C.P. 04510, Mexico; 2Instituto Nacional de Psiquiatría Ramón de la Fuente Muñiz, Mexico City C.P. 14370, Mexico

**Keywords:** risk-taking, recreational domain, tower building task, Balloon Analogue Risk Task, Sensation Seeking Scale

## Abstract

Risk-taking is a fundamental aspect of life spanning diverse contexts. Despite many tests being readily available, the behavioral assessment of risk propensity in recreational contexts, where decisions are not necessarily motivated by monetary gains or losses, is not well represented. As the tower building task (TBT) shares features of recreational activities, we evaluated whether it could assess risk-taking in this type of scenario. In the TBT, participants use standard-size wooden blocks to build the tallest tower they can within a 10 min period. In the current study we (i) examined methodological modifications to the task to either promote or reduce risky behavior and (ii) tested possible associations between the TBT scores and those of two commonly used risk-taking evaluations: The Balloon Analogue Risk Task (BART) and the Sensation Seeking Scale (SSS). We found that limiting the number of permitted collapses decreased the willingness to take risks, whereas challenging participants to exceed a “record” height increased risk-taking. TBT scores of one of the conditions correlated with scores of the BART and the SSS, particularly with the subscale addressing recreational activities. These findings suggest that the TBT offers a potentially useful means of evaluating risk-taking behaviors akin to those of the recreational domain.

## 1. Introduction

Risk-taking can be defined as behavior that involves a potential loss while providing the opportunity to obtain a reward [1]. Negative consequences can include financial loss, disappointment or physical harm, while positive consequences can include winning money, excitement or well-being. Risk propensity within one context does not guarantee the same risk propensity in another; for example, someone can be risk averse when it comes to a gamble but risk prone in situations involving their own safety and health [2]. This illustrates how risk tendencies vary across contexts and highlights the need to aggregate risk-taking behaviors into categories or domains (e.g., [3,4,5]).

Evidence of the difficulty of measuring risk-taking can be inferred from the large number of evaluation methods. Most risk propensity assessments are based on self-reports of prototypical risky behaviors (e.g., reckless driving, unprotected sex) or risk-related constructs, such as sensation seeking [6,7] and impulsivity [8]. Observational studies (e.g., [9]), epidemiological data (e.g., [10]) and economic games (e.g., [11]) have also been used. Additionally, various laboratory-based behavioral tasks have been developed to simulate realistic risk-taking within a controlled scenario. They mostly entail consecutive trials in each of which the participant must choose between a gamble and a safe option. Examples include the array of levers of [12] in which participants can accrue rewards by pulling levers, but also may lose their cumulative gain if they reach the “disaster lever”. Similarly, in the Balloon Analogue Risk Task (BART) [13] the participant inflates virtual balloons; each pump accrues a monetary reward, which is lost if the balloon bursts.

Although there is an ample array of behavioral tools with which risk-taking can be evaluated, the majority focus on the assessment of decisions whose consequences involve numeric losses as well as numeric gains, as happens with gambling and with financial investments. Such assessments constrain behavior to point events, such as pressing a button, drawing a card from a deck or pulling a lever. This facilitates the application of consecutive trials and eliminates the influence of skill on risk-taking. Moreover, in these experiments, participants are usually motivated extrinsically by means of monetary or edible rewards.

However, not all risky behaviors manifest in such a binary, black and white manner. A prime example is recreational risk-taking. Recreation is a common human activity, to which many people dedicate considerable time and resources, and which is mainly driven by intrinsic motivation. In such activities, pleasure and satisfaction are derived from performing the activity itself [14], which contrasts with extrinsic incentives, such as earning money, avoiding punishment, or complying with social norms. Another characteristic of this domain is that decision making is partially tied to skill or perceived skill, that is, participants’ performance influences the outcome (e.g., [15]). Moreover, in recreational activities negative consequences vary in terms of magnitude (e.g., from a slight scratch to losing a limb); thus, individuals’ responses can be graded according to the feedback received during the activity. Finally, recreational rules do not usually favor rigidity in behavior, but promote diverse approaches in order to attain the goal.

Here, we test whether the recently developed tower building task (TBT) [16] is a useful tool to help fill this gap, that is, the behavioral evaluation of risk-taking in a recreational context. In this task, participants are instructed to use wooden blocks to build the tallest tower they can within a limited time. The TBT is a harmless and ludic activity that incorporates risk-taking behaviors with features akin to those of recreational contexts, including: (i) the challenge to build a tall tower is driven by intrinsic motivation, as it is rewarding in its own right [17], (ii) skill adds to the manner and speed in which blocks are placed, (iii) participants can adjust their behavior during the construction process based on feedback within a single trial, and (iv) an experimental design that permits diverse ways to achieve a tall tower.

In the present study we tested two methodological modifications to the TBT aimed at encouraging either risk prone or risk averse attitudes towards the building process while maintaining the intrinsic motivation and the ludic nature of the task. Participants were also evaluated using two other risk-assessment tools: the BART and the Sensation Seeking Scale (SSS). Our first aim was to evaluate whether the modifications to the TBT had an effect on participants’ behavior and our second aim was to identify possible associations between the measures of risk-taking on the TBT and those derived from the other two forms of risk propensity assessment.

## 2. Materials and Methods

### 2.1. Participants

We recruited 120 undergraduates aged 18–26 years (50% self-identified as men, 50% as women; M = 21.26, SD = 2.10) attending a public university in Mexico City. Participants were recruited using flyers placed on message boards on campus or by invitation directly in their classrooms. Students could schedule an appointment with the experimenter via WhatsApp. Individuals were informed that the study focused on human decision-making within a ludic and harmless context and that depending on the task (see below) they could receive a monetary reward based on performance. Participants with visible or self-reported motor impairments were allowed to participate but their results were not included in the analysis.

### 2.2. Assessment Tools

#### 2.2.1. Tower Building Task (TBT)

The task consisted in a single participant building the tallest tower she or he could using wooden blocks (1.5 × 2.5 × 7.5 cm, Figure 1a) from the board game Jenga (Parker Brothers, Hasbro Inc, Beverly, Massachusetts, USA) within 10 min. A set of blocks was placed next to the participant’s dominant hand (Figure 1b). Each had a maximum of 108 blocks to build the tower and were allowed to remove blocks from the current effort and to replace them as they wished. They were instructed to build on a 50 × 50 cm smooth melamine board (Figure 1c). An hourglass was placed beside the board to inform them of the time remaining (Figure 1d) without performing numeric calculations, just as happens in many recreational scenarios. They were told they could stop at any moment if satisfied with their tower. Thus, using up the allotted time or all blocks was not mandatory.

##### TBT Conditions

We tested 120 participants in one of four variants of the TBT (Table 1), resulting in four independent groups (n = 30, all self-identified as sexually binary: 50% men, 50% women). Methodological modifications from the original version [16] included testing a single participant as well as adding (i) a record as an incentive to encourage participants to take risks and (ii) limiting efforts to a single collapse so as to promote a risk-averse attitude.

A verbatim version of the instructions per condition (in Spanish) is available in Appendix A. Criteria to consider the involuntary loss of height as a collapse as well as an explanation of the record reference used are given in Appendix A, respectively.

##### Primary Dependent Measure of the TBT: Fixed Height Gain

Three variables (height, number of blocks added, and duration of the trial) were used to calculate the fixed height gain. Tower height was measured by adding the height of the pieces making up the tower from a screenshot taken immediately before the outcome (either a collapse, ending the task early or after the 10 min had elapsed). The total of blocks and the exact duration of the task was obtained through video coding (see below).

The fixed height gain is the increase in height (cm) per piece added adjusted according to the duration of the trial or (height of tower/number of pieces) * proportion of the trial. Height gain values were corrected by time, as the value of height gain taken on its own cannot differentiate between a participant who chooses to build a short tower (e.g., 18 cm with 6 pieces giving a height gain of 3) and quickly chooses to stop building, from that of a participant who chooses to build a tall tower (e.g., 90 cm with 30 pieces also giving a height gain of 3) but took the whole time of the trial. Clearly the second option carries greater risk as the increase in height is achieved at the expense of the tower’s stability. In addition, adjusting for the duration of the task is necessary as the trial can come to an end because the participant (i) keeps building until the end of the trial, (ii) chooses to finish before the allotted time, (iii) or the trial ends because of a collapse (only for SC and RSC conditions).

#### 2.2.2. The Balloon Analogue Risk Task (BART)

The BART [13] is a computer-based tool which provides a context in which risk propensity is assessed. The participant can earn a monetary reward by inflating simulated balloons that are shown (one at a time) on a computer screen. On the right bottom of the screen, the earnings which correspond to the balloon that is being inflated are shown. Whenever the participant presses the spacebar, the balloon inflates and consequently, a certain amount of money is added to the temporary reserve. In the current study each click (i.e., pump) entailed an increase in size (3 cm in all directions) while rewarding the participant with MXN 0.05 (1 MXN~0.05 USD). If the balloon was inflated past its limit and burst, the accrued money from that balloon was lost. Each time the balloon burst it produced a sound. During each trial the participant could decide when to stop pumping (by pressing the enter button) to prevent the balloon from bursting, and consequently keep the money accumulated. If so, the money accrued in the temporary bank was transferred to the permanent account. Participants were given 20 trials, which is considered acceptable as studies with 30 trials produce similar results [18].

As suggested by Lejuez [13], the primary dependent measure for risk propensity assessment was the “adjusted” average number of pumps, or the number of pumps for balloons that did not burst, that is, the point at which the participant made an active choice to keep the money and move to the next balloon.

#### 2.2.3. The Sensation Seeking Scale (SSS)

The SSS [7] is a self-administered questionnaire of 40 items. Each item consists of a forced choice between two opposing statements related to the willingness to engage in novel, diverse and intense experiences. Each item in which the choice is the “sensation seeking” option is summed to produce an overall score which ranges from 0 to 40. Higher scores suggest higher levels of sensation seeking that, according to Zuckerman et al. [7], entail “the need for varied, novel, and complex sensations and experiences and the willingness to take physical and social risks for the sake of such experience”. Items can be grouped into one of four constructs: (i) Thrill and Adventure Seeking, (ii) Experience Seeking, (iii) Disinhibition, and (iv) Boredom Susceptibility. The internal consistency of the SSS is good, with Cronbach’s alpha coefficients ranging from 0.83 to 0.86 [7,19]. Here, we used a Spanish language version of the questionnaire from a translation made by the authors (see Appendix A).

### 2.3. Procedure

Participants received instructions describing the general purpose of the study and a brief description of the tasks they were about to undertake; in particular, they were informed that a cash reward would be paid depending on their performance on the BART. All agreed to participate in the study, after which each was randomly allocated to one of the four conditions of the TBT until 30 participants (15 men and 15 women) had been tested in each condition. Testing took place between October 2019 and February 2020.

Each participant was tested individually in a single 1 hr session that included all three evaluations. First, they performed the TBT, then the BART, and then answered the SSS. Tasks were presented always in the same order so as to match the ascending degree of risk explicitness expressed in the instructions: in the TBT, the participant’s choices could lead to a negative outcome (e.g., tower collapsing) but had no impact on monetary reward; in the BART, gambling with the potential monetary reward emphasized the task’s risk-taking nature with monetary consequences, and the SSS included several choices explicitly related to risk-taking (e.g., “I can’t understand people who risk their necks climbing mountains”). Since risk-taking was not explicit on the TBT (i.e., not mentioned or implied in the instructions or testing phase) participants were not influenced to adopt a certain risk attitude to comply with any expectation. With this, we attempted to prevent biasing risk behavior.

Each task took place in different unoccupied rooms. For the TBT, a video camera mounted on an adjustable tripod filmed the building board from 2 m away. The participant was asked to sit beside the board and a set of 108 blocks was placed next to their dominant hand. Additionally, a 10-min hourglass was placed beside the board. For the conditions involving a record reference, a wooden post was placed next to the board with a mark on it representing the record height to be exceeded. Participants were instructed to build the tallest tower possible within ten minutes using the wooden blocks that were provided while ensuring that the tower was built on the board. Participants were told they could use all blocks or just a part of the set, and that they could remove and put back blocks into the structure. Using a model set of blocks, the experimenter explicitly demonstrated the three possible positions in which the blocks could be laid (i.e., the vertical and the two horizontal forms). Importantly, participants were reminded they could spend the entire trial building or that they could finish earlier if they were satisfied with the tower they had built. Attempts were unlimited except for the two conditions (SC and RSC) in which the collapse of the tower ended the trial. The experimenter remained in the room in all the TBT sessions to manage the video camera, make observations and clarify any doubts.

The BART was then administered. The following instructions were given: A series of 20 red balloons will be presented across the task. You should increase the size of the balloons one at a time by clicking the spacebar. Each click represents a pump, which in turn increases your profit. The larger you inflate the balloon, the greater amount of money you will accrue in a temporary bank, but if the balloon bursts you will lose the money for that balloon and will move on to the next one of the series. In order to prevent losing the reward you can keep the money from the inflated balloon whenever you decide to by pressing enter (the monetary reward goes to a permanent bank) and automatically the next balloon of the series will come up on the screen. Be aware that some balloons burst earlier than others, meaning that the size that can be reached varies; in fact, some balloons may cover the entire screen.

Following the BART, participants completed the SSS. Although instructions were clearly written at the top of the questionnaire’s first page, the experimenter read them out loud, pointing out that the questionnaire involved 40 forced-choice items. In addition, we underlined the fact that answers would not be regarded as either correct or incorrect, to emphasize that participants should reply honestly and choose the statement that better fitted their behavior.

For both the BART and the SSS, the experimenter remained nearby in case of technical doubts. After all tasks were performed, participants received their monetary reward and were asked not to share information about the contents of the tasks; although we could not control how well they complied with this request. Three participants were excluded (two women and one man) due to equipment failure, resulting in a final sample size of 117 participants.

The recruitment process and the experimental procedures met the bioethical requirements established by the Internal Review Board for Research with Human Subjects of the Instituto de Investigaciones Biomédicas, UNAM.

### 2.4. Behavioral Coding

Video files of participants working on the TBT were analyzed using a free event logging software [20] to code the sequence of events within each trial encompassing block additions, tower collapses, and duration of the trial. With these we calculated participants’ fixed height gain. A trained second rater scored these same metrics on randomly selected videos (n = 24; 20%). Interrater reliability was calculated using a Spearman correlation. Values were significant and equal to or above *r* > 0.98. Participants’ performance on the BART was automatically transcribed into a separate spreadsheet for each participant. As the SSS was a paper-and-pencil-based self-administered questionnaire, we transcribed participants’ responses for each item into a spreadsheet that contained all answers. The sequence of events describing the building procedure of each participant, their performance on the BART and their SSS scores can be accessed at: https://doi.org/10.6084/m9.figshare.16654618.v1 (accessed on 19 of July 2022).

### 2.5. Statistical Analysis

Since our data were not normally distributed according to the results of Shapiro–Wilk tests, we evaluated differences between TBT conditions using a Kruskal–Wallis test and calculated effect size as *E*^2^ [21]. This was followed by post hoc Dunn tests for pairwise multiple comparisons, which are appropriate for groups with an unequal number of observations [22]; to adjust *p*-values for multiple comparisons we used the Holm procedure. Additionally, we performed Spearman rank correlations between the main dependent measures of the TBT and the BART (i.e., fixed height gain vs. adjusted average number of pumps) as well as of the TBT and the SSS (i.e., fixed height gain vs. sensation seeking score). Additional Spearman rank correlations were also used to test the association between fixed height gain on the TBT and each of the SSS subscales (Thrill and Adventure Seeking, Experience Seeking, Disinhibition, Boredom Susceptibility). Statistical analyses were done using R [23] and all plots were built using the package *ggplot2* [24]. Statistical significance was set at *p* < 0.05.

## 3. Results

### 3.1. Comparison among Conditions of the TBT

Regarding our first aim, we found that limiting the number of collapses and/or implementing a reference-record had an effect on the participants’ behavior (Figure 2). Significant differences were found among the TBT conditions for fixed height gain (*H*(3) = 26.44, *p* < 0.0001, *E*^2^ = 0.23), which was significantly smaller in both the SC and the RSC conditions when each was compared to the R condition (SC vs. R, *p* < 0.001; R vs. RSC, *p* < 0.001). Additionally, fixed height gain for the SC condition was significantly smaller compared to that of the BL condition (BL vs. SC, *p* < 0.05). We did not find gender differences in any of the conditions. See scores of additional descriptors of behavior for each TBT condition in Appendix A.

### 3.2. Correlations of Risk-Related Metrics between TBT and the Other Tools of Risk Assessment

Considering the degree of association between the fixed height gain of each of the TBT conditions and the corresponding adjusted average number of pumps on the BART, we found a significant, positive moderate-to-strong correlation for the SC condition (*r* = 0.46, *p* = 0.01; Figure 3a), but not for the remaining conditions (*r* = 0.02 for BL, *r* = −0.07 for R, and *r* = 0.22 for RSC). Considering the degree of association between the fixed height gain of each of the TBT conditions and the total score on the SSS, we found a significant, positive moderate-to-strong correlation for the SC condition (*r* = 0.47, *p* = 0.009; Figure 3b), but not for the remaining conditions (*r* = −0.28 for BL, *r* = −0.14 for R, and *r* = 0.11 for RSC). See scatter plots of the non-significant associations between fixed height gain and the two other risk-related measures in Appendix A. Moreover, a similar correlation value was found between the fixed height gain from the SC condition and the score yielded from the Thrill and Adventure Seeking subscale (*r* = 0.43, *p* = 0.02; Figure 3c), one of the four subscales of the SSS. No significant correlations were found for the other subscales.

## 4. Discussion

In the present study, we evaluated methodological modifications of the TBT designed to either increase participants’ risk propensity or promote a risk averse attitude by implementing four conditions. For each of the four conditions we also correlated participants’ performance with that of two other risk assessment tools: the BART and the SSS. The main findings were (i) that the modifications to the TBT had the expected effect on participants’ risk-taking behavior, and (ii) that fixed height gain, our main risk-related behavioral dependent measure, in the SC condition was associated with the measures of risk-taking in the BART and SSS.

Regarding the methodological modifications of the TBT, we found that limiting the trial to one attempt appeared to induce an aversion to loss as suggested by the lower fixed height gain in the SC condition. Participants’ apparent aversion to loss could have been the result either of overweighing the probability of loss, which is what risk-averse decision makers do [25,26,27], or simply because a subjectively greater value was allocated to the tower built when only one attempt was permitted. The implementation of a record, mainly when attempts were unlimited, seemingly had the opposite effect: it encouraged participants to exhibit a greater risk propensity. This might have been due to a priming effect of the visually signaled record, or because the stated record implied competitiveness. Risk-taking and competitiveness are particularly hard to disentangle, as both are core aspects of recreational activities. Some authors [28,29] have approached the evaluation of risk-taking through gambling and/or investment games and competitiveness by means of activities, such as tossing a tennis ball into a bucket or rope jumping. In this sense, employing the TBT in future studies could show how competitive interactions (either in-person or by means of a record) influence risk propensity. For instance, athletes might display an increased risk propensity than non-athletes when evaluated in a competitive situation. Taken together, the present findings show that even with a modest number of participants, performance on the TBT was sufficiently sensitive to show differences in risk-taking behavior among independent groups tested under different conditions.

We found a positive and significant correlation between fixed height gain in the SC condition of the TBT and the adjusted average number of pumps on the BART. The fixed height gain reflects participants’ willingness to increase the height of the tower at the expense of the tower’s stability in a similar way to which the adjusted average number of pumps reflects participants’ willingness to increase the size of the balloon while increasing the chances of making it burst and consequently losing the reward. These measures share at least two features. First, each successive gain (i.e., block addition or pump) increases the amount to be lost (i.e., height or cash) in case of a negative outcome (i.e., a collapse or burst). Second, both cases involve a decrease in the relative gain (i.e., the relative gain becomes smaller with each block addition or successive pump). These two features resemble real-life situations involving risks which, as Lejuez [13] indicates, often result in diminishing returns with an increased potential of experiencing negative consequences.

We also found a significant correlation between fixed height gain in the SC condition with the total score of the SSS. The association was not only with the overall score but specifically with the Thrill and Adventure Seeking subscale, which mainly includes questions related to recreational activities. This supports the notion that the TBT offers a relevant behavioral measure that may predict, to some extent, risk attitudes that manifest in the recreational domain. One explanation is that the willingness to increase the height of the tower by either placing vertical pieces in the structure or increasing the speed of addition, carries a thrilling sensation as the pursued and seemingly valued outcome is under threat of collapse. Few studies have focused on a behavioral approach attempting to measure risk-taking based on activities that compromise safety and where some thrilling sensation probably emerges, e.g., [15,30].

Despite the similarities mentioned above, some important distinctions highlight why the TBT could be considered a tool for evaluating risk-taking in a recreational context. In the TBT, decisions may affect the outcome in a non-linear manner (e.g., a block can be added in diverse ways, resulting in gains of height and/or stability, or in a collapse) much as happens in recreational activities (e.g., a step up may bring us closer to the summit, provide better grounding, or result in falling down a cliff) supporting the idea that the nature of the task promotes diverse approaches to reach the goal. Similarly, building approaches on the TBT allow for skill-modulated responses (e.g., adjusting piece placement if the tower wobbles) just as happens in recreational activities (e.g., decreasing climbing speed in bad weather). Furthermore, in contrast with self-report methods, the TBT may be considered applicable to almost any human group, regardless of income or cultural background. Given the breadth and complexity of risk-taking domains, alternatives such as the TBT could usefully augment the array of behavioral tools for evaluating risk-taking by offering a safe and ludic task appropriate to the recreational domain.

### Limitations and Perspectives

Some limitations of the current study can be acknowledged. Tasks were presented in a fixed order so as to minimize participants’ tendency to adopt a certain attitude towards risk; this potential bias could be addressed in the future by counterbalancing task presentation. We did not evaluate participants’ visuospatial abilities or fine motor skills; although both are factors that may influence building performance. Furthermore, participants’ academic course type was not included as a variable due to the modest sample size. Although calculating fixed height gain scores may be too laborious for clinical contexts, small aids, such as a tally counter for the number of pieces added, a grid in the background to estimate the height of the tower, and a chronometer for task duration, could help simplify assessment. Additionally, the sample was restricted to university students, who may arguably be homogenous in their risk propensity. Thus, future studies should consider testing a more diverse sample.

A main aim of the present study was to test different variants of the TBT to identify the best experimental setup for future testing. In this sense, it may be that a single collapse could be too harsh a condition in samples including very young or impulsive participants whose towers collapse very soon into the test, which may not reflect their actual willingness to increase height per piece added and where some adjustment to the test protocol might be needed. In such cases, we may consider allowing participants to continue building after their tower collapses, while still stating at the start that only one attempt is permitted so that they are unaware of the possibility for a subsequent attempt. Additionally, further testing of the association between the TBT and scales evaluating risk-taking in specific domains is also needed (e.g., [3,31]).

## 5. Conclusions

Given that risk attitudes vary across domains it is necessary to address risk propensity using a multimethod approach. The current study describes and evaluates the TBT, whose main measure, the fixed height gain, is associated with measures of risk propensity assessment on other widely used tasks. As the TBT involves the potential for modulated skill-related responses, diverse strategies for pursuing the goal and an apparent intrinsic motivation, it seems to be a potentially useful measure of risk behaviors akin to those in recreational activities. Thus, we propose it as a novel behavioral test with which to explore the recreational domain.

## Figures and Tables

**Figure 1 behavsci-12-00325-f001:**
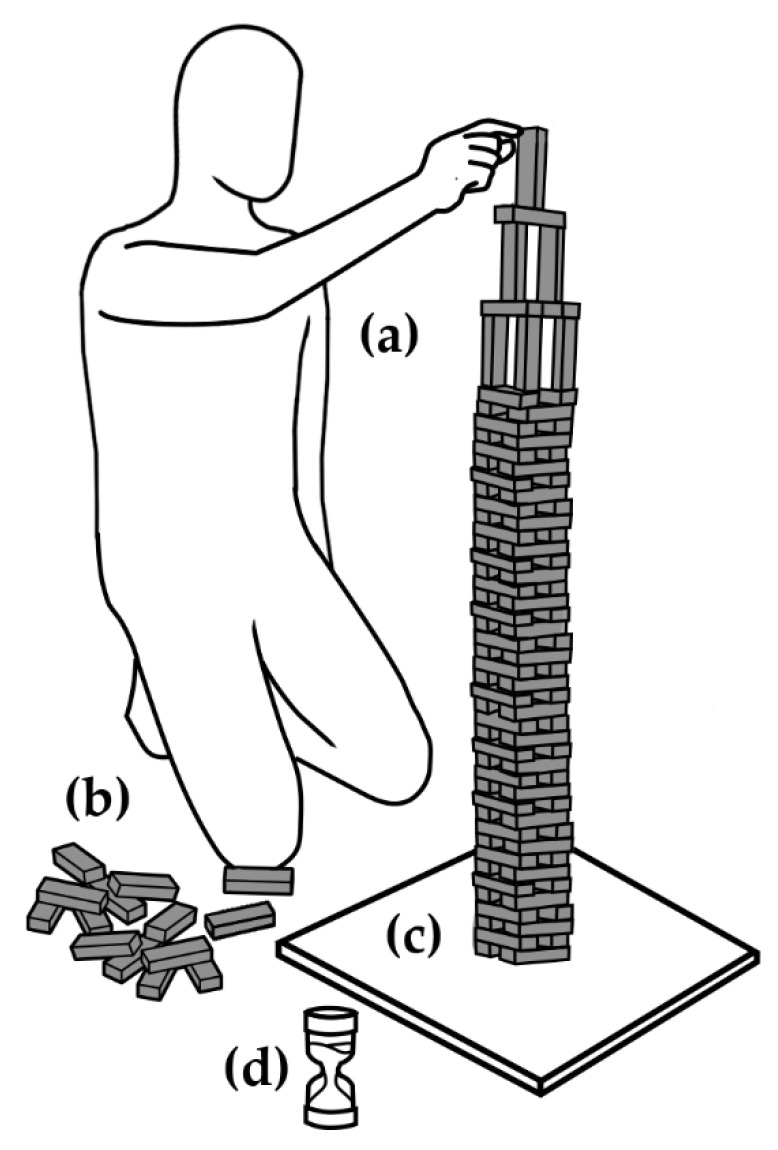
Experimental setup. The participant was instructed to build the tallest tower he or she could (**a**) using standard size wooden blocks (**b**) over a flat, uniform melamine surface (**c**) before time on the hourglass ran out (**d**). See text for additional instructions that varied per experimental condition.

**Figure 2 behavsci-12-00325-f002:**
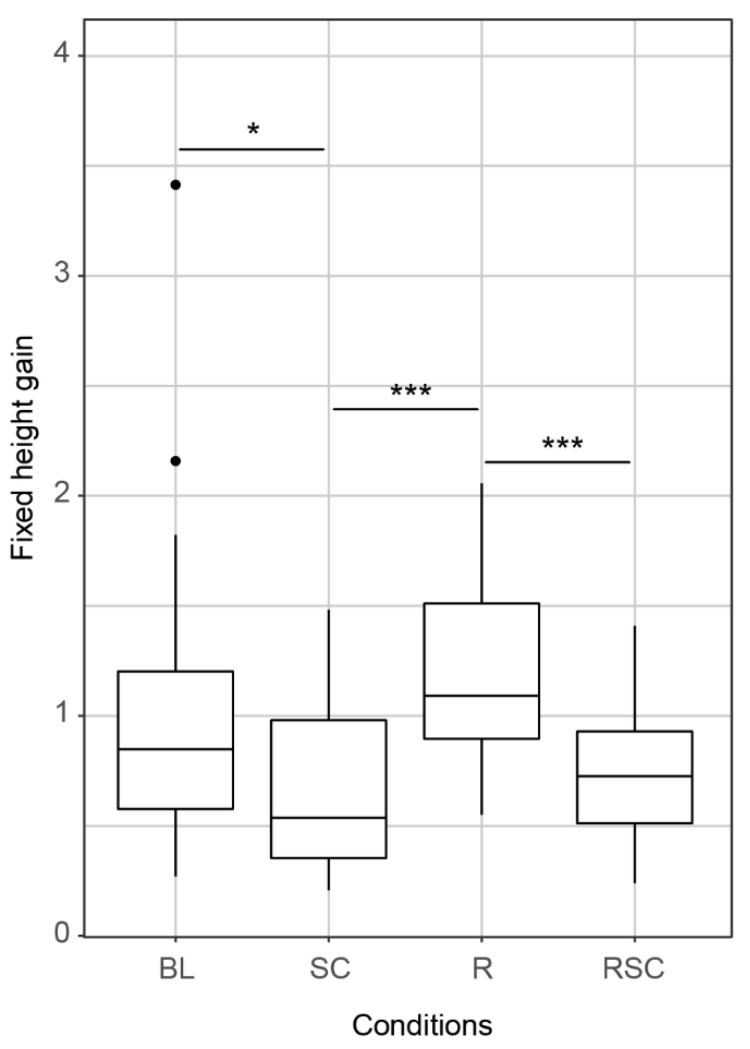
Boxplots of fixed height gain on the TBT (i.e., an index that reflects the height gained per block added adjusted according to the duration of the trial) per condition. The x axis gives the different conditions: Baseline (BL), Single Collapse (SC), Record (R), and Record-Single Collapse (RSC). Horizontal lines through the boxes give median values, box limits represent the 1st and 3rd quartiles. Whiskers extend from the limits of the boxes to the smallest and largest values no further than 1.5 times the interquartile range. Outliers (filled circles) lay beyond this range. Horizontal bars across the top of boxplots indicate statistical significance between two conditions. Asterisks represent levels of significance: * *p* < 0.05; *** *p* < 0.001.

**Figure 3 behavsci-12-00325-f003:**
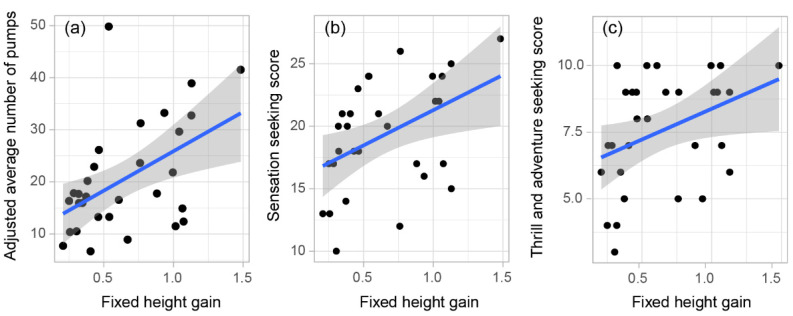
Scatter plots for the Single Collapse condition of the TBT in relation to (**a**) the BART, (**b**) the SSS, and (**c**) the Thrill and Adventure Seeking subscale of the SSS. Filled circles show participants’ fixed height gain on the *x*-axis, and on the *y*-axis (**a**) the adjusted average number of pumps, (**b**) the overall SSS, and (**c**) the SSS subscale Thrill and Adventure Seeking score. Blue lines show the linear association between the two variables of each plot and the gray areas indicate the confidence intervals.

**Table 1 behavsci-12-00325-t001:** Methodological modifications resulting in four variants.

	No Record	Record
**No collapse limit**	Baseline (BL) We expected an increased risk propensity compared to SC, and a decreased risk propensity compared to R.	Record (R) We expected an increased risk propensity in comparison to the other three conditions.
**Trial ends if tower collapses**	Single Collapse (SC) We expected decreased risk propensity in comparison to the other three conditions.	Record-Single Collapse (RSC) We expected an intermediate risk propensity: greater than SC but lower than R.

## Data Availability

Data reported in this paper are available at https://doi.org/10.6084/m9.figshare.16654618.v1 (accessed on 7 July 2022).

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
