# Peer review of "The Tower Building Task: A Behavioral Tool to Evaluate Recreational Risk-Taking"

_behavsci, 2022, doi:10.3390/bs12090325_

Round 1
Reviewer 1 Report
The paper has a novel and interesting topic which is presented well and clearly structured.
There are just a few points which I would raise for the authors consideration:
1) In the section 'TBT conditions', it is not clear of the time period for the testing. Did it take place over a day, a week, were the other tests also on the same day? I would ask the authors to consider adding this to give the reader a clearer picture, although the experimental process is very well presented in its current form. This question is also important in terms of whether groups / individuals had the opportunity to communicate with others about the tests. How did the authors ensure the testing wasn't tainted by students telling each other what to expect / give tips.
2) The choice of sample could be argued as affecting the results and yet it was not considered as a potential limitation / future research direction to consider a different sample. Arguably, students have less to lose, and age is a factor. Likewise, risk taking has been considering from the aspect of gender and course type. Have the authors considered the data according to these splits?
3) A record as an incentive, to me, seems to appeal to a person's competitive instinct or desire to appear successful. Were these aspects considered or could this be a direction for future research?
4) The sequence of testing was TBT, BART and then SSS. Is there some reason for this? It would be worth adding some reasoning for the choice, and as in the first point, whether these took place on the same day for all groups. Consider also if the choice of sequencing might have affected the findings.
Author Response
We appreciate the reviewers’ comments and suggestions. These helped us clarify some ambiguous points and expand on relevant aspects of the discussion that have improved the manuscript. Below you can find our response to each of the reviewers’ queries in italics. Line numbers are provided in each of these responses so that you can find the additions (in red) in the corrected version of the manuscript.
Reviewer #1
In the section 'TBT conditions', it is not clear of the time period for the testing. Did it take place over a day, a week, were the other tests also on the same day?
I would ask the authors to consider adding this to give the reader a clearer picture, although the experimental process is very well presented in its current form. This question is also important in terms of whether groups / individuals had the opportunity to communicate with others about the tests. How did the authors ensure the testing wasn't tainted by students telling each other what to expect / give tips?
The choice of sample could be argued as affecting the results and yet it was not considered as a potential limitation / future research direction to consider a different sample. Arguably, students have less to lose, and age is a factor. Likewise, risk taking has been considering from the aspect of gender and course type. Have the authors considered the data according to these splits?
R >> We appreciate the reviewer’s attention to detail. We now emphasize the reason behind this choice in lines 195-196. We have also included how this choice might affect the findings in the Limitations and perspectives section (lines 376-378).
Reviewer 2 Report
The paper is well organized with proper structure. The bibliography is sufficient and well given.
Specifically, the technical terms are explained in detail and the topic of the paper is clear and understandable.
The presented methodology and the results are clearly communicated, with the necessary background for the readers included in the paper.
The review of the state-of-the-art is sufficient. It includes references to other relevant studies that have been previously proposed for the discovery of relations.
The novel contribution of the paper is highlighted, as well.
The conclusion section includes a discussion about the results obtained by this work, but it doesn't demonstrate previous works on the analysis of the same or similar data.
Author Response
We appreciate the reviewers’ comments and suggestions. These helped us clarify some ambiguous points and expand on relevant aspects of the discussion that have improved the manuscript. Below you can find our response to each of the reviewers’ queries in italics. Line numbers are provided in each of these responses so that you can find the additions (in red) in the corrected version of the manuscript.
Reviewer #2
The conclusion section includes a discussion about the results obtained by this work, but it doesn't demonstrate previous works on the analysis of the same or similar data.